

# Investigating the Response of Land-Atmosphere Interactions and Feedbacks to Spatial Representation of Irrigation in a Coupled Modeling Framework

Patricia Lawston-Parker[1,2], Joseph A. Santanello Jr.[2], Nathaniel W. Chaney[3]

[1]Earth System Science Interdisciplinary Center, University of Maryland, College Park, MD, 20740, United States
[2]Hydrological Sciences Laboratory, NASA Goddard Space Flight Center, Greenbelt, MD, 20771, United States
[3]Department of Civil and Environmental Engineering, Duke University, Durham, NC, 27708, United States

*Correspondence to*: Patricia Parker (patricia.m.parker@nasa.gov)

**Abstract.** The characteristics of the land surface play a critical role in determining the transfer of water, heat, and momentum to the atmosphere. Together with the model physics, parameterization schemes, and parameters employed, land datasets determine the spatial variability in land surface states (i.e., soil moisture and temperature) and fluxes. Despite the importance of these datasets, they are often chosen out of convenience or regional limitations without due assessment of their impacts on model results. Irrigation is an anthropogenic form of land heterogeneity that has been shown to alter the

land surface energy balance, ambient weather, and local circulations. As such, irrigation schemes are becoming more prevalent in weather and climate models with rapid developments in dataset availability and parameterization scheme complexity. Thus, to address pragmatic issues related to modeling irrigation, this study uses a high-resolution, regional coupled modeling system to investigate the impacts of irrigation dataset selection on land-atmosphere (L-A) coupling using a case study from the Great Plains Irrigation Experiment (GRAINEX) field campaign. The simulations are assessed in the

context of irrigated versus non-irrigated regions, subregions across the irrigation gradient, and sub-grid scale process representation in coarser scale models. The results show that L-A coupling is sensitive to the choice of irrigation dataset and resolution and that the irrigation impact on surface fluxes and near surface meteorology can be dominant, conditioned on the details of the irrigation map (i.e., boundaries, heterogeneity, etc), or minimal. A consistent finding across several analyses was that even a low percentage of irrigation fraction can have significant local and downstream atmospheric impacts,

suggesting that representation of boundaries and heterogeneous areas within irrigated regions is particularly important for the modeling of irrigation impacts on the atmosphere in this model. When viewing the simulations presented here as a proxy for 'ideal' tiling in a Earth System Model scale gridbox, the results show that some 'tiles' will reach critical nonlinear moisture and PBL thresholds that could be important for clouds and convection, implying that heterogeneity resulting from irrigation should be taken into consideration in new sub-grid land-atmosphere exchange parameterizations.



## 1 Introduction

The characteristics of the land surface play a critical role in determining the transfer of water, heat, and momentum to the atmosphere (Chaney et al. 2018; Santanello et a. 2018; Zhou et al. 2019; Pielke Sr. 2001). For this reason, an important
component of Earth System Models is the land model, which represents the radiative and physical properties of the surface, providing a lower boundary for, and exchange with, the atmosphere (Peters-Lidard et al. 2015). Datasets that define the land surface and its spatial variability (i.e. land heterogeneity), such as land use and land cover (LULC), soil properties (e.g., type and texture), and vegetation characteristics (e.g., leaf area index, greenness vegetation fraction) are often overlooked, but are integral components of this land surface representation. Together with the model physics, parameterization schemes, and
parameters employed, these datasets determine the spatial variability in land surface states (i.e., soil moisture and temperature) and water and energy balance via land surface temperature and fluxes of latent and sensible heat (Niu et al. 2011; Yang, et al. 2011). Despite the importance of these datasets, they are often chosen out of convenience or regional limitations without due assessment of their impacts on model results.

Operational and global weather and climate models, such as ESMs, tend to operate at relatively coarse scales compared to
the natural variability of the land surface. As a result, advanced approaches to representing sub-grid scale heterogeneity of the land have been introduced (e.g., tiling), but not fully leveraged due to model coupling that primarily exchanges a gridscale mean flux between land and atmospheric models (Simon et al. 2021). To address these deficiencies in current operational Earth System Models (ESMs), NOAA's Climate Process Team (CPT) and the Coupling of Land and Atmospheric Subgrid Parameterizations (CLASP) project (http://www.clasp.earth) seeks to improve the parameterization of
heterogeneous sub-grid exchange between the land and atmosphere. Ideally, such a parameterization should be representative of both natural (e.g. land cover, soil type, terrain, etc) and human-induced (e.g. irrigation, reservoirs, dams, etc.) sources of heterogeneity.

With respect to the latter, modeling of the geophysical impacts of human activities is a relatively new area of research. In particular, agricultural irrigation consumes the largest amount of water by far at the global level (FAO, 2021) and has been
shown to alter the land surface energy balance, ambient weather, and local circulations (Bonfils and Lobell, 2007; Lo and Famiglietti 2013; Rappin et al. 2022). As such, irrigation schemes are becoming more prevalent in weather and climate models with rapid developments in dataset availability and parameterization scheme complexity (e.g, Valmassoi et al. 2020; Zhang et al. 2020; X. Xu et al. 2019; Lawston et al. 2015; Leng et al. 2013). In most regional and global models, an irrigation fraction map is used to determine where irrigation can activate, and together with the triggering algorithm and
thresholds, creates unique spatial variability in soil moisture that alters the naturally (i.e., from precipitation alone) occurring heterogeneity. Until recently, the choice of irrigation fraction dataset was limited, but increasing availability of datasets (e.g., Deines et al. 2019; Brown and Pervez 2014; Siebert et al. 2013) has created a pressing need to better understand how land-atmosphere (L-A) coupling responds to different spatial representations of irrigation. Such an investigation is relevant not only for future coupled modeling of irrigation (e.g., in terms of implications/limitations for scientific results), but also for



understanding where and when such irrigation-imposed heterogeneity may be important for subgrid parameterizations, such as those being developed in the CLASP project.

Thus, to address pragmatic issues related to modeling irrigation, this study uses a high-resolution, regional coupled modeling system to investigate the impacts of irrigation dataset selection on land-atmosphere (L-A) coupling. The results are discussed in the context of ESM subgrid heterogeneity to better understand how L-A coupling may be impacted by developments in

CLASP parameterizations. However, the results are relevant to the overall modeling community that is rapidly working towards developing approaches to parameterize irrigation. The main questions this work seeks to answer are 1) What is the impact of irrigation dataset selection on land surface heterogeneity in soil moisture and surface fluxes? 2) How does irrigation induced heterogeneity impact L-A interactions and feedbacks at the 1 km, process level scale? 3) Is there essential land-atmosphere coupling information that is lost when averaging from the process level scale (1 km) to the scale of a typical

ESM (e.g., 100 km)? The paper is organized as follows:  Section 2 presents relevant background information, Section 3 describes the methods and experimental design, including the modeling systems and observations employed for this work. Results are given in Section 4, with discussions and conclusions presented in Sections 5 and 6, respectively.

## 2 Background

The characteristics and spatial variability (i.e., heterogeneity) of the land surface directly affect the surface energy and

moisture budgets (Chaney et al. 2018, 2021; Zhou et al. 2019) and therefore play a key role in simulation and prediction of the atmosphere. Previous work has shown that landscape heterogeneity influences the spatial structure of surface heating, convective initiation, and cumulus cloud base height (Rabin et al., 1990; Schrieber et al. 1996; Pielke Sr. 2001; Tian et al. 2022). Recent studies have assessed the relative importance of common sources (i.e., datasets) of land heterogeneity in LSMs and coupled models for a range of applications. For example, Simon et al. (2021) showed that the land heterogeneity

which produces the biggest impacts on clouds and mesoscale circulations in the Weather Research and Forecasting (WRF) model in large eddy simulation (LES) mode is primarily driven by heterogeneous meteorological forcing (i.e. precipitation). In addition, Li et al. (2022) found that including more land heterogeneity sources in the E3SM earth system model led to larger spatial variability in the simulated water and energy partitioning, with atmospheric forcing and land use land cover sources contributing the most.

Irrigation is a form of anthropogenic land heterogeneity that increases soil moisture, and therefore has the potential to affect ambient weather via alterations to the surface energy and water budgets and planetary boundary layer (PBL) feedbacks. Many previous studies have concluded that irrigation can repartition latent and sensible fluxes, ultimately resulting in local to regional irrigation-induced cooling (Aegerter et al. 2017; Leng et al. 2017; Qian et al. 2013; Mahmood et al. 2013; Lawston et al. 2020). Other studies have found that irrigation can generate new circulations or modify those that already

exist. For example, Harding and Snyder (2012) found that irrigation enhances precipitation but also leads to a net water loss in the U.S. Great Plains as the precipitation falls away from the source and is often outweighed by ET increases. While Lo





and Famiglietti (2013) also showed that irrigation strengthens the regional hydrological cycle through increased ET and water vapor export, some of the additional water is returned to the area via streamflow and managed diversion. Mahalov et al. 2016 found that irrigation modifies the North American Monsoon rainfall such that some areas downwind experience

increases in convective rainfall through positive soil-moisture rainfall feedbacks, while other area experiences decrease in precipitation due in part to decreased CAPE. These studies, and others focused on the impact of irrigation on weather and climate (Kang and Eltahir 2018; Thiery et al. 2017; Cook. et al. 2010), demonstrate that irrigation can have large impacts on near surface meteorology, PBL evolution, mesoscale circulations, and convective initiation.

To comprehensively observe the impacts of irrigation on the atmosphere, the Great Plains Irrigation Experiment

(GRAINEX; Rappin et al. 2021) deployed a collection of observing systems in a 100 x 100 km region of eastern Nebraska in May – August 2018. This field campaign, funded by NSF, was centered on a divide between predominately irrigated (west) and rainfed agriculture (east). Observing systems used during the campaign include 12 flux towers, 80 temporary meteorological observing stations, two vertical wind profilers, and regular radiosonde launches (Rappin et al. 2021). The campaign also featured two intensive observation periods (IOPs), 1) 29 May – 13 June, and 2) 16-30 July, to more rigorously

observe the impacts of the commencement and peak of irrigation, respectively. Analysis of the GRAINEX data has shown that air temperature, wind speed, and PBLH were lower over the irrigated area as compared to the non-irrigated region (Rappin et al. 2021, 2022) and that irrigation in the upslope region of the domain weakened terrain-induced baroclinicity and the slope wind circulation (Phillips et al. 2022).

The land heterogeneity imposed by irrigation is a result of by human behavior, including local and regional water

management policies that can influence farmer decisions regarding crop types, timing, and water use. To simulate irrigation, a model approximates such behavior by progressing through a series of checkpoints to determine: 1) where, 2) when, and 3) how much irrigation water to apply. The activation of the irrigation scheme (i.e., 'when') and 'how much' water is applied can be prescribed on a schedule (e.g., Valmassoi et al. 2020) or conditioned on a model variable, most often soil moisture, meeting a predetermined threshold of dryness and desired replenishment (e.g., Ozdogan et al. 2010; Lawston et al. 2015).

Of primary importance is 'where' irrigation is triggered in the model, as it is the prerequisite to determining the details of irrigation in 2) and 3) above. In regional and global models, maps of irrigated areas are processed into irrigation fraction maps that define the fraction of the model gridcell that is irrigated. These irrigation fraction maps are used to establish where (spatially) in the model domain the irrigation scheme 'can' activate and may also be referenced to scale the amount of water applied (e.g., Ozdogan et al. 2010, Lawston et al. 2020, Nie et al. 2021). Many modern irrigation maps are created by

leveraging the geophysical impacts of human behaviors, as observed by remote sensing platforms, sometimes combined with survey statistics or climate data, to create maps of areas equipped for irrigation (Siebert et al. 2013) or actual irrigated areas (Thenkabail, 2006; Biggs et al., 2006; Ozdogan and Gutman, 2008; Brown and Pervez 2014; Salmon et al. 2015; Deines et al. 2019). Although once prohibitively difficult to acquire at high temporal frequency, technological advances in tools (e.g., Google Earth Engine, machine learning algorithms) and computing power have increased the availability of irrigation

datasets (Deines et al. 2019; T. Xu et al. 2019).



One of the first datasets to leverage machine learning applications in remote sensing used an image classification algorithm along with MODIS vegetation indices, ancillary climate and agricultural data to map irrigated areas in the Continental United States circa 2001 (hereafter MOD; Ozdogan and Gutman, 2008). The resulting dataset, produced at 500 m resolution, showed an estimated RMSE of about 2% of total irrigated area in the U.S. Using a similar methodology, but expanding the analysis to the global scale, The Global Rainfed and Irrigated Paddy Croplands (GRIPC, Salmon et al. 2015) also used a machine learning algorithm applied to MODIS data, climate data, and existing information from other datasets, to map not only irrigated areas, but also rainfed and paddy croplands at 500 m, circa 2005. More recently, Deines et al. (2019) used Google Earth Engine to process Landsat data and environmental covariables using a Random Forecast classifier to create the Annual Mapping of Irrigated Areas – High Plains Aquifer (AIM-HPA) dataset, consisting of one map per year from 1984-2017 at 30 m resolution for the High Plains Aquifer region of the Central U.S. The high temporal and spatial resolution of this dataset marks a major advancement in irrigation mapping that has benefits not only for weather and climate modeling, but also the management, policy, and agronomy fields (Deines et al. 2019).

This work explores the impacts of these three, high quality and widely-used irrigation datasets; MOD, GRIPC, and AIM-HPA (Table 1) on land-atmosphere interactions in eastern Nebraska using a case study from the GRAINEX field campaign. It should be noted that the purpose of this study is not to discern the most accurate irrigation map for the study area. Rather, this work seeks to understand if and to what extent irrigation heterogeneity (via irrigation map selection and resolution) can impact simulation and prediction of land atmosphere coupling and ambient weather, and discusses the implications of such impacts in the context of sub-grid scale process representation in coarser scale models.

## 3 Methods

### 3.1 Models and Experimental Design

This study uses the Noah land surface model version 3.3 (Chen and Dudhia, 2001) within the NASA Land Information System (LIS; Kumar et al. 2006) to complete long term (2010-2019), land-only spinups of land surface states (soil moisture and temperature) and fluxes (sensible and latent). A long term LSM spinup that is consistent in its irrigation treatment is essential to proper, equilibrated initialization of the subsequent coupling simulations. The modeling domain is 360 x 360 km with a spatial resolution of 1 km resolution and encompasses the GRAINEX field campaign study region (Fig. 1). The land-only simulations are forced with meteorological data from the National Land Data Assimilation System-Phase 2 (NLDAS-2; Xia et al. 2012) and use MODIS International Geosphere-Biosphere Program (MODIS-IGBP) landcover and NCEP climatological greenness vegetation fraction (GVF) and leaf area index (LAI) datasets.

The irrigation parameterization within LIS/Noah is activated when four conditions are met: 1) The land cover must be an irrigable type (i.e., not urban, bare soil, etc), 2), The irrigation fraction must be non-zero, 3) The simulation date/time must be within the 'growing season', defined by a gridcell GVF greater than 40% of the annual range in climatological GVF, and





4) The root zone must be dry enough to require irrigation, as determined by root zone moisture availability that falls below 50% of field capacity (Ozdogan et al. 2010; Lawston et al. 2017). The root zone is determined by the crop type and scaled by
GVF to mimic a seasonal cycle of root growth. If all conditions are met, water will be applied as precipitation (mimicking a 'sprinkler' application) until the root zone moisture availability reaches 80% of field capacity. The irrigation fraction is used to scale the amount of water applied. More details and evaluation of the irrigation scheme can be found in Ozdogan et al. (2010) and Lawston et al. (2017). The irrigation scheme, thresholds, and all datasets except irrigation fraction (i.e., land cover, GVF, soil texture, crop type, meteorological forcing) are kept constant between runs.

Three different irrigation maps, MOD, GRIPC, and AIM-HPA (see Section 2) are used in the land-only simulations. These datasets have relatively high native resolution (30 m for AIM-HPA and 500 m for MOD and GRIPC), but are upscaled to the model 1 km grid at varying resolution to discern not only the impact of differing irrigation sources, but also of varying dataset resolution. The AIM-HPA dataset for the year 2017 (the most recent year available at the time of this work) is upscaled to both 1 km and 12 km, GRIPC is upscaled to 1 km, and MOD is upscaled the resolution of the NLDAS domain
(~12.5 km). These simulations provide the opportunity to discern the impacts of heterogeneity resulting from the following: 1) single dataset resolution only (i.e., 1km AIM-HPA vs. 12km AIM-HPA), 2) different datasets at coarser resolution (MOD (12km) vs. 12km HPA) and higher resolutions (i.e., GRIPC (1km) v. 1km HPA), and 3) Using a product at an 'off-the-shelf' resolution (i.e., MOD (~12km)). An additional baseline run with the irrigation scheme inactive is also completed (hereafter NO IRR), resulting in five runs (i.e., NO IRR, MOD 12km, GRIPC 1km, AIM-HPA 1km, AIM-HPA 12km).

Each land-only spinup is used to initialize a fully coupled simulation using the NASA Unified Weather Research and Forecasting System (NU-WRF; Peters-Lidard et al. 2015), which is coupled to LIS (i.e., LIS-WRF; Kumar et al. 2008). These 5 LIS-WRF simulations are run for 30 hours from 00 UTC 24 July to 12 UTC July 25, 2018. This date was selected to build upon previous GRAINEX analyses (Rappin et al. 2021, 2022) that identified this day as the most ideal day for the evaluation of irrigation's impact on land-atmosphere coupling during the second IOP. The LIS-WRF simulations are forced
using meteorological data from the NCEP Final Analysis (FNL; https://rda.ucar.edu/datasets/ds083.3/). The model setup has 60 vertical levels and uses Mellor-Yamada-Nakanishi-Niino level 2.5 (MYNN2.5; Nakanishi and Niino 2006) surface layer and PBL schemes. The identical irrigation scheme, thresholds, and respective datasets used in the land-only simulations are also used in the coupled runs, ensuring continuity between the spun-up initial conditions and the coupled irrigation parameterization.

Figure 1 shows the irrigation maps applied to the modeling domain for the 4 irrigated runs along with markers indicating the locations of relevant GRAINEX observation sites and boxes defining subregions used in the analysis detailed in Section 4. There are several key differences that emerge among the maps. When comparing the higher resolution maps (i.e., Fig 1b vs. Fig 1d), AIM-HPA extends the irrigated area farther to the east and south than the GRIPC dataset and has a wider range of irrigated fraction values. That is, most gridcells classified as irrigated by GRIPC show a high irrigation fraction (>90%)
while AIM-HPA is more heterogeneous, even in the heart of the irrigated area. Some of this heterogeneity is lost when upscaling to 12 km (Fig. 1c), but a greater range in irrigation fraction still exists as compared to the other coarser resolution



run, MOD (Fig 1e). The location of highest intensity irrigation in the MOD dataset is similar to the others (i.e., roughly between 97-98°W), but MOD is an outlier in that it extends a low percentage of irrigation (i.e., <30%) far eastward, well into what is considered to be rainfed areas of the GRAINEX study domain.

## 3.2 Observations

Observations from the GRAINEX field campaign are used to assess the model simulations. A full description of all instruments deployed during the campaign can be found in Rappin et al. 2021. Figure 2 shows the locations of the comprehensive land and PBL profiling instruments used in this study, overlaid on irrigation fraction given by the AIM-HPA dataset. Specifically, we use temperature and humidity data from the Environmental Monitoring, Economical Sensor Hubs (EMESH) stations (green, orange, red circles), latent and sensible heat fluxes from 12 flux towers (yellow squares), and radiosonde observations from the Roger's Farm and York Integrated Sounding System (ISS) sites (blue triangles). In order to investigate the irrigation dataset heterogeneity impacts in different subregions of the domain, the EMESH stations are classified as being irrigated (green circles), transition (orange circles), or rainfed (red circles), as discussed further in Section 4.

## 3.3 L-A Interactions

The local land atmosphere coupling (i.e., LoCo; Santanello et al. 2018) process chain paradigm provides an integrative framework for assessing the impacts of land surface heterogeneity (LSH) by assessing the relative sensitivities of 1) surface fluxes to soil moisture, 2) PBL evolution to fluxes, 3) entrainment fluxes to PBL evolution, and 4) the collective feedback of the atmosphere on ambient weather, clouds, and precipitation. This allows a more comprehensive analysis of the coupling impacts of heterogeneity versus a traditional one-at-a-time approach (e.g. evaluating evapotranspiration, or air temperature independently). In this study, the complete set of process chain variables are not available at any individual site, so we do not undertake a site by site, end-to-end LoCo assessment. Rather, we use the process chain framework to assess the bulk land surface forcing by using aggregates of observations and models across regions. In particular, we employ 1) Evaporative Fraction (EF = latent heat flux/(latent heat flux + sensible heat flux)) vs. PBL Height (PBLH) plots, and 2) a modified version of mixing diagrams (Santanello et al. 2009; 2018). Traditional mixing diagrams relate the diurnal co-evolution of temperature and moisture in the boundary layer along with vectors representing the surface input of heat and moisture scaled by the PBLH. As surface fluxes and PBLH observations are not co-located with near surface meteorology observations, the vectors are excluded from this analysis.

## 4 Results

Figure 3 shows the differences between the no irrigation and each irrigation simulation in top layer soil moisture, EF, temperature at 20 m, humidity (mixing ratio) at 20 m, and PBL height, along with the GRAINEX instrument locations for



reference. The 20 m model height (rather than 2 m) is chosen throughout the analysis to assess the bulk (ambient) signal of irrigation, as 2 m values will be more reflective of hyper-local vegetation and crop characteristics and irrigation practices in the field. In each case, irrigation increases soil moisture, latent heat flux, and near surface humidity, as expected, with a

corresponding decrease in sensible heat flux, temperature, and PBL height. The spatial pattern of changes in soil moisture and evaporative fraction corresponds closely with the irrigation data source as the irrigation map determines the location of soil moisture increases, which then directly affects fluxes via the terrestrial leg of the L-A coupling chain (Dirmeyer 2011). The impacts on PBL height and temperature and humidity are more representative of the bulk atmospheric, integrated (spatially and vertically) response to irrigation (i.e. the atmospheric leg of L-A coupling) and, as such, the impacts extend a

bit beyond the actual boundaries of irrigated area, particularly in the southern region as result of a light northerly winds.

The spatial extent and magnitude of irrigation-induced changes varies based on the selected irrigation map, with the biggest differences stemming from dataset source, rather than resolution. The MOD map is the clear outlier as the irrigated area, and subsequent impacts extend well east into the actual rainfed region of the GRAINEX domain. The biggest changes are in the southeast corner of the domain where even a small amount of irrigation fraction (4-16%) increases soil moisture by 0.1-0.15

$m^3$ $m^{-3}$ and reduces temperature by up to 3K. The GRIPC map most closely matches the GRAINEX site-level classification of irrigated and rainfed sites, largely bisecting the site locations and limiting most impacts to Nebraska, while the others extend south into Kansas. The AIM-HPA maps at different resolutions are quite similar, but the upscaling limits the precision of the spatial heterogeneity at the irrigated vs. non-irrigated border, and subsequent impacts.

In order to analyze the extent to which essential L-A coupling information, driven by the irrigation heterogeneity, is retained

at the scale of ESM, three 100 x 100 km subregions are imposed on the study domain, indicated by the boxes in Fig. 1. The western subregion is largely irrigated, the center contains the transition area from irrigated to rainfed, and the eastern is predominantly rainfed. These subregions can be viewed as a proxy for a ESM gridcell, with the model 1 km gridcells contained within them serving as 'tiles' in which the ESM subgrid land-atmosphere processes are fully resolved.

Figure 4 presents plots of daytime average EF versus maximum PBLH, which is a critical coupling metric that relates the

daytime surface flux of heat and moisture (i.e. the land forcing) to the PBL response (Santanello et al. 2009; 2011). The gray circles represent the EF and PBLH (given as the model diagnostic from the MYNN scheme) values for each 1km gridcell within the irrigated subregion for each model run. The colored circles show the subregional average for each model run and therefore are the same for each subfigure (a-e). The NO IRR run (Fig 4a) shows the variability in EF and PBLH that results from natural heterogeneity in the model (i.e., landcover, soil type, etc) and reveals that this subregion is fairly wet on this day

even without irrigation, as most gridcells have EF values of 0.5 or greater. In the irrigated runs (Fig b-e), the irrigation scheme increases EF in most gridcells and greatly reduces the PBLH height as compared to NO IRR. Although the spatial averages are very similar across the irrigated runs (e.g., colored markers grouped near 0.8 (EF), 1100 m (PBLH)), the spread across the 1km 'tiles' varies considerably. For example, the AIM-HPA 1km dataset produces the most variability in EF, due to the local heterogeneity that extends to lower and, importantly, zero irrigation fraction values. This variability in EF is lost

when upscaling to 12km (Fig. c). In fact, the AIM-HPA 12km more closely resembles the MOD 12km run (Fig 4d), which



has positive irrigation fraction throughout the region, than the AIM-HPA 1km run, suggesting that resolution of the irrigation fraction dataset can play a key role in the terrestrial leg of L-A coupling. Despite the lower EF values seen in this subregion in AIM-HPA and GRIPC runs, there is little impact on PBL growth, likely due to the bulk of the domain being irrigated and the spatial and vertical blending of that influence.

Figure 5 shows EF versus PBLH plots for the transition region for all runs. The NO IRR run (Fig. 5a) shows tight grouping of points with fairly clean borders governed by the natural heterogeneity (e.g.,landcover, soil type, vegetation characteristics, etc) and associated model thresholds and parameters. The irrigated runs (Fig 5 b-e) show how irrigation, and also the irrigation fraction map itself, change the heterogeneity in EF v. PBLH. For example, in the transition region, the MOD run extends irrigation to the east such that the 'transition' region is almost entirely irrigated, resulting in EF values that are

skewed towards the high (i.e., wet) end of EF, while in the AIM-HPA 1km run, two clusters (wet vs. moderate) emerge as a result of the precisely resolved irrigation boundary. Notably, the wetter cluster in the AIM-HPA 1km run has EF values similar to the MOD run (i.e., ~0.65-0.9), but has corresponding PBLH values that are up to 400 m higher than in MOD. It is likely that the drier (east of the AIM-HPA transition boundary) gridcells that have higher sensible and lower latent heat impact larger PBL growth and entrainment feedbacks (as the PBL can integrate over 10-50 km horizontally; Stull 1988), and

the influence is felt beyond just the non-irrigated region. This implies that some gridcells in the AIM-HPA 1km run (but not MOD, which has irrigation throughout) reach critical moisture and PBL thresholds that allow for PBL feedbacks that increase the height of the PBL. The L-A interactions that lead to these feedbacks are a direct result of the irrigation map and triggering thresholds and, in turn, could be important for cloud development and convective processes in a subgrid ESM parameterization.

The rainfed region (Fig. S1) shows considerable spread in both EF and PBLH but little differences across runs as all datasets specify zero, or near-zero, irrigation fraction. This allows 'natural' heterogeneity in EF to dominate, which is dependent on land cover, soil type, terrain, and precipitation. Overall, it is clear from the intercomparison of the three subregions that irrigation (and choice of dataset in coupled models) can be a dominant and/or limiting (Fig. 4bc), conditional (dependent on the orientation of the fraction map, boundaries, and wind flow; Fig. 4de), or minimal (i.e. natural heterogeneity dominates;

Fig. 4fg) control on L-A coupling and the processes that govern the relationship of soil moisture, fluxes, PBL growth on ambient weather.

Figure 6 presents mixing diagrams for 7 subregions of the GRAINEX domain. All irrigation maps define the western part of the domain as heavily irrigated, but the maps differ considerably in their representation of the heterogeneity *within* the irrigated area and in defining the location and characteristics of the transition between irrigated and rainfed areas. To address

this 'within region' heterogeneity, the EMESH sites are classified using the AIM-HPA 1 km irrigation map into 7 different sub-regions: North, Middle, and South Irrigated (Fig. 2, green circles); Northeast, Northwest, and Southwest Transition (Fig. 2, orange circles); and Rainfed (Fig. 2, red circles). Table 2 lists the specific site numbers included in each subregion.

Figure 6 shows that in the Middle (Fig. 6b) and South (Fig. 6c) Irrigated regions there is tight grouping across (i.e., minimal difference in) the irrigation runs. Irrigation can affect the simulations through 1) wetter initial soil moisture conditions from





the irrigated spinup (i.e., 'previous irrigation'), and 2) irrigation in the coupled simulation (i.e., 'present irrigation'). The similar performance displayed by the irrigation runs is due to the combined facts that 1) there is agreement among the maps that these regions are heavily irrigated, and 2) these regions also exhibit low antecedent moisture (Fig. S2), causing irrigation to turn on in the coupled run. In contrast, the North (Fig. 6a) Irrigated region, while also heavily irrigated, has wetter antecedent soil moisture, so irrigation doesn't turn on in the coupled run at the sites that make up this region. Rather, the
spread is due to previously applied irrigation from which the model has begun to dry out.

The irrigation maps differ the most in the transition region between irrigated and rainfed, leading to the greatest spread in model runs in the Northwest (Fig. 6e) and Southwest (Fig. 6f) transition regions. In the rainfed (Fig. 6g) region, the MOD run is the outlier as it classifies the rainfed sites as irrigated, while all other runs are close to the NO IRR run as they correctly classify these sites as rainfed. Figure 6 also shows that the model has an inherent dry bias, as the observations are
consistently more humid than the NO IRR run in all subregions, including rainfed. The irrigation scheme, regardless of prescribed irrigation fraction map, acts to mitigate this bias, compensating for other model errors beyond only the lack of irrigation in the model.

In order to better understand the irrigation impacts to the PBL, the York (irrigated) and Rogers Farm (rainfed) sites are analyzed in more detail in Figs. 7-9. Figure 7 again shows mixing diagrams, but for the single sites (i.e., not spatial averages
as in Fig. 6) of York and Rogers Farm, using temperature and humidity at 20 m in the model and at the level closest to 20 m from radiosonde observations. At the York site, the model starts the day considerably cooler and drier than observations, but the daytime heating vigorously warms and moistens the boundary layer, bringing all model runs closer to observations by late morning (e.g. 15 UTC). The model follows the diurnal cycle of the observations well, implying that it captures the warming and moistening of the boundary layer early in the day, followed by drying due to entrainment midday (i.e., the 'left
turn' in the diagram), then the second round of moistening late in the day (i.e., the final 'right turn').

At the Rogers Farm site, most of the model runs capture the midday drying well (slow leftward curve), but miss the small morning moistening from 13-15 UTC shown in observations. In addition, the models display a late day moistening (20-23 UTC or 3-6 pm local) that does not appear in observations. The exception is the MOD run, which shows the early day moistening and gradual drying throughout the day, more consistent with observations. The better performance of MOD is
due to misclassifying this site as a small percentage irrigated, and therefore mitigating the existing dry bias in the model.

Figures 8 and 9 show two-hourly (15 UTC 24 July to 01 UTC 25 July) potential temperature profiles for the lowest 1.8 km at the York and Roger's Farm sites, respectively. A detailed analysis of the radiosonde profiles is available in Rappin et al. 2021. Here, we focus on the differences in the model runs and their ability to simulate what was observed. At the York site, the observations show more rapid PBL growth from 15 to 17 UTC than the model as the PBL grows into a more unstable
free atmosphere layer. Although the model is slower in simulating this growth at 17 UTC, the well mixed layer and PBLH (i.e., approximated as the level corresponding with the maximum gradient in potential temperature) are remarkably well simulated during 19-23 UTC. The NO IRR simulation is the outlier, as it simulates a warmer boundary layer throughout the diurnal period. Little difference is noted between the irrigated runs at this site.



At the rainfed site (Fig. 9), the model runs show a warm bias in the potential temperature profiles. The OBS are again more
unstable in the free atmosphere and feature a residual layer extending to about 1300 m at 15 UTC. The observed PBL grows
quickly to the top of the residual layer and the PBLH reaches a maximum of about 1400 m at 19 UTC. The model simulates
a shallower residual layer (about 900 m at 15 UTC) and slower growth once the PBL surpasses the top of the residual layer
and grows into the more stable air in the free atmosphere. Thus, these plots illustrate that the residual layer can be a good
predictor of future PBL growth (e.g. Santanello and Friedl 2005) and that surface changes induced by irrigation need to be
considered holistically, as the PBL and lower troposphere are important modulators of the response. Figure 9 also shows
more sensitivity to irrigation fraction map (i.e., spread among runs) at the Rogers Farm site as compared to York, despite
irrigation not occurring at this site, except in the misclassified MOD.

Figures 10 and 11 show water vapor mixing ratio profiles for the York and Rogers farm sites, respectively. At the York site,
the model again exhibits a dry bias that irrigation acts to erode as it moves the irrigation runs slightly closer to observations.
There is little difference between the runs early in the day while the AIM-HPA runs at each resolution perform marginally
better by 21 and 23 UTC, but all of the runs perform quite well. The observations at both sites show a dry layer around 1500
m that gradually lowers throughout the day. At the rainfed site (Fig. 11), the MOD run, which exhibits greater moisture from
the irrigated misclassification, lacks this dry layer, resulting in a wetter and taller boundary layer that stands out from the
other runs. Overall, the models struggle to simulate the details of the elevated dry slot with the MOD run outlier as a result of
the irrigation misclassification.

## 5 Discussion and Conclusions

This study used a high resolution regional coupled modeling system to assess the impacts of the spatial representation of
irrigation on L-A coupling using a case study from the GRAINEX field campaign. The simulations are assessed in the
context of irrigated versus non-irrigated regions, subregions across the irrigation gradient, and sub-grid scale process
representation in coarser scale models.

The results show that L-A coupling is sensitive to the choice of irrigation dataset and resolution and that the irrigation impact
on surface fluxes and near surface meteorology can be dominant, conditioned on the details of the irrigation map (i.e.,
boundaries, heterogeneity, etc), or minimal. For example, within the irrigated region, irrigation map resolution had a larger
influence on the spatial heterogeneity of evaporative fraction than choice of dataset, while the opposite is true (i.e., dataset
was more important) in the transition region. When viewing the simulations presented here as a proxy for 'ideal' tiling in a
ESM scale gridbox, the results show that some 'tiles' will reach critical nonlinear moisture and PBL thresholds that could be
important for clouds and convection, implying that heterogeneity resulting from irrigation should be taken into consideration
in new sub-grid land-atmosphere exchange parameterizations, such as those being investigated under the CLASP project.

A consistent finding across several analyses was that even a low percentage of irrigation fraction can have significant local
and downstream atmospheric impacts, suggesting that representation of boundaries and heterogeneous areas within irrigated



regions is particularly important for the modeling of irrigation impacts on the atmosphere. In addition, analysis of modeled and observed temperature and moisture profiles demonstrated that lower troposphere stability is an important modulator of the irrigation signal. The results also show that irrigation, regardless of dataset, acts to mitigate an existing dry bias in the model, as highlighted by irrigation improving the bias in rainfed areas misclassified by the irrigation datasets. This

underscores that care must be taken in the implementation of irrigation physics in models to avoid utilizing the irrigation scheme as a tuning mechanism to compensate for embedded model errors. Approaching the evaluation of irrigation schemes through a holistic, L-A coupling framework, as demonstrated here, can aid in disentangling model improvements that result from new irrigation inclusion versus mitigation of unrelated model biases.

This study focused on a major type of human induced land heterogeneity (i.e., irrigation) that can be introduced by model

parameters and datasets, distinct from natural sources such as land use land cover, soil properties (type and texture), and greenness vegetation fraction. Our results suggest that the combination of irrigation fraction specification and triggering algorithm creates a new type of soil moisture heterogeneity that is different from what would occur due to atmospheric forcing alone, and results in changes to L-A coupling and ambient weather.

The selected case study was chosen to fully leverage the available GRAINEX data on a favorable day for land-atmosphere

interactions during IOP2 (i.e., the height of irrigation) and to build on previous GRAINEX and L-A coupling work (Rappin et al. 2021, 2022). The model antecedent soil moisture in the irrigated region ranged from 0.14 to 0.24 $m^3$ $m^{-3}$ as some regions of the domain were wet from antecedent rainfall. In addition, only one type of LSM irrigation scheme and thresholds are used in this study, where previous work has shown that L-A coupling is sensitive to irrigation type and factors such as vegetation greenness (Lawston et al. 2015). This enabled a controlled study on the impacts of the specific irrigation map applied, which is often overlooked in irrigation modeling impact studies (similar to land cover or soil type datasets). The

combination of mapping, thresholds, and antecedent soil moisture regime (which influences triggering) determine the irrigation heterogeneity, and deserve further investigation under a wider range atmospheric conditions. In addition, this analysis concentrated on the region of GRAINEX observations, where there are other areas of the domain with larger differences between datasets (e.g., the southeast corner) that could be explored more directly.

This work featured only three, of several, widely-used irrigation datasets, and it is likely that the amount and variety of irrigation datasets will increase in coming years. The results of this study suggest that to be most beneficial for irrigation representation within earth system models, future irrigation fraction datasets should ideally be high resolution, resolve large scale irrigation boundaries, and capture within irrigation heterogeneity. While not investigated here, a focus of future work should be to discern the importance of the location and intensity of irrigation map boundaries on local wind and mesoscale

circulations, as well as prevailing wind influences across these boundaries.



## Code/Data Availability

NASA's Land Information System code is open source and available on GitHub (https://github.com/NASA-LIS/LISF). Data from the GRAINEX field campaign is accessible via https://data.eol.ucar.edu/master_lists/generated/grainex/. NASA Unified WRF code and model results are archived and available upon request.

## Author Contribution

PLP and JAS formulated the experimental design. PLP completed the experiments, analyzed the simulation results and field campaign data. JAS and NWC aided in interpretation and implications of resutls. PLP prepared the manuscript with contributions from all co-authors.

## Competing Interests

The authors declare that they have no conflicts of interest.

## Acknowledgements

This work was supported by NOAA Climate Process Teams - Translating Land Process Understanding to Improve Climate Models Grant # NOAA-OAR-CPO-2019-2005530. Resources supporting this work were provided by the NASA High-End Computing (HEC) Program through the NASA Center for Climate Simulation (NCCS) at Goddard Space Flight Center.

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

| Dataset Abbreviation | Dataset Name | Reference | Resolutions used |
|---|---|---|---|
| AIM-HPA | Annual Irrigation Maps – High Plains Aquifer | Deines et al. 2019 | 1 km and 12 km |
| GRIPC | Global Rainfed and Irrigated Paddy Croplands | Salmon et al. 2015 | 1 km |
| MOD | MODIS-based dataset | Ozdogan and Gutman, 2008 | ~12.5 km |

**Table 1. List of irrigation maps used in the simulations, including references and resolution.**






| Site Number | Classification | MOD (12km) | GRIPC (1km) | AIM-HPA (1km) | AIM-HPA (12km) |
|---|---|---|---|---|---|
| 95 | North Irrigated | 46 | 100 | 66.2 | 77.5 |
| 96 | North Irrigated | 46 | 100 | 79.1 | 82.2 |
| 51 | North Irrigated | 41 | 50 | 95.2 | 86.3 |
| 50 | North Irrigated | 22 | 55.6 | 86 | 80.5 |
| 98 | North Irrigated | 58 | 100 | 55.5 | 83.4 |
| 23 | Middle Irrigated | 65 | 100 | 75.4 | 85 |
| 54 | Middle Irrigated | 61 | 100 | 80.9 | 85 |
| 13 | Middle Irrigated | 44 | 50 | 14.8 | 53.8 |
| 26 | Middle Irrigated | 54 | 100 | 50.2 | 74.2 |
| 67 | Middle Irrigated | 43 | 100 | 30 | 74.9 |
| 9 | Middle Irrigated | 59 | 100 | 56.2 | 80.7 |
| 57 | South Irrigated | 47 | 100 | 97.2 | 85.5 |
| 68 | South Irrigated | 45 | 25 | 36.5 | 60.6 |
| 25 | South Irrigated | 18 | 83.3 | 84 | 60.6 |
| 52 | South Irrigated | 44 | 16.7 | 61.7 | 64.8 |
| 69 | South Irrigated | 26 | 83.3 | 90.1 | 77.3 |
| 81 | South Irrigated | 15 | 0 | 92.3 | 47.2 |
| 34 | South Irrigated | 38 | 100 | 79.6 | 70 |
| 53 | Northwest Transition | 3 | 0 | 0 | 0 |
| 38 | Northwest | 4 | 0 | 0 | 1.2 |





| | | | | | |
|---|---|---|---|---|---|
| | Transition | | | | |
| 99 | Northwest Transition | 5 | 0 | 0 | 3.7 |
| 71 | Southwest Transition | 7 | 0 | 8.7 | 36.1 |
| 85 | Southwest Transition | 7 | 0 | 84.7 | 36.1 |
| 66 | Southwest Transition | 7 | 0 | 9.8 | 36.1 |
| 70 | Southwest Transition | 9 | 100 | 35.7 | 39 |
| 83 | Southwest Transition | 12 | 50 | 79.1 | 39 |
| 84 | Southwest Transition | 14 | 50 | 6.6 | 44 |
| 86 | Southwest Transition | 7 | 0 | 16.2 | 57.4 |
| 42 | Northeast Transition | 52 | 0 | 33.3 | 55 |
| 48 | Northeast Transition | 30 | 0 | 62.8 | 24.9 |
| 47 | Northeast Transition | 0 | 0 | 84.3 | 58.6 |
| 40 | Northeast Transition | 4 | 0 | 73.7 | 52.3 |
| 39 | Northeast Transition | 0 | 0 | 85.4 | 46.4 |
| 41 | Northeast Transition | 6 | 0 | 0 | 9.2 |
| 5 | Rainfed | 0 | 0 | 0 | 0 |
| 8 | Rainfed | 4 | 0 | 0 | 0 |
| 11 | Rainfed | 6 | 0 | 0 | 0 |

**Table 2. List of GRAINEX EMESH sites, including the subregional classification used in Fig. 6, and the irrigation fraction given by each dataset for the gridcell closest to the site latitude and longitude.**




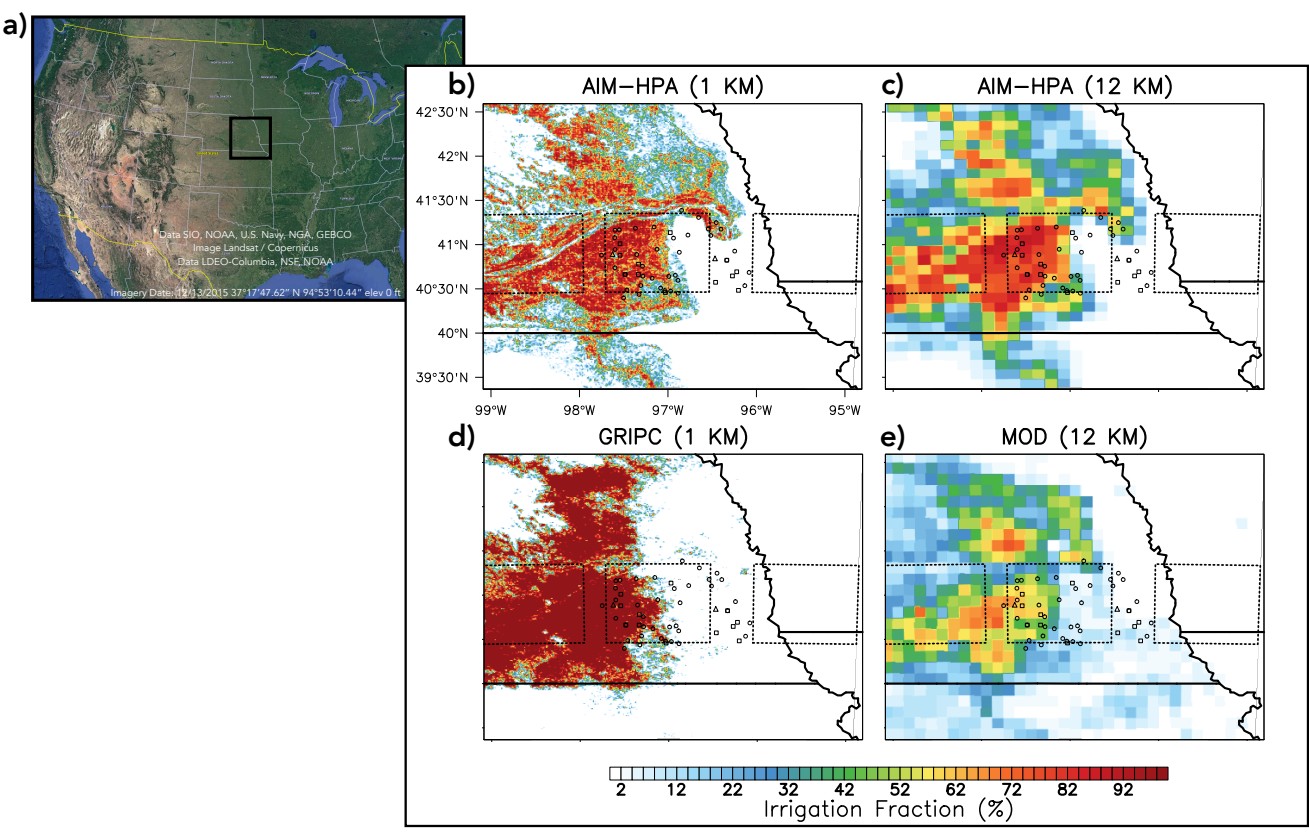

Figure 1: a) Map of the United States (Google Earth Pro, 2023) with box indicating the location of the study area. The b) AIM-HPA 1km, c) AIM-HPA 12km, d) GRIPC 1 km, and e) MOD 12 km irrigation fraction datasets applied to the modeling domain.



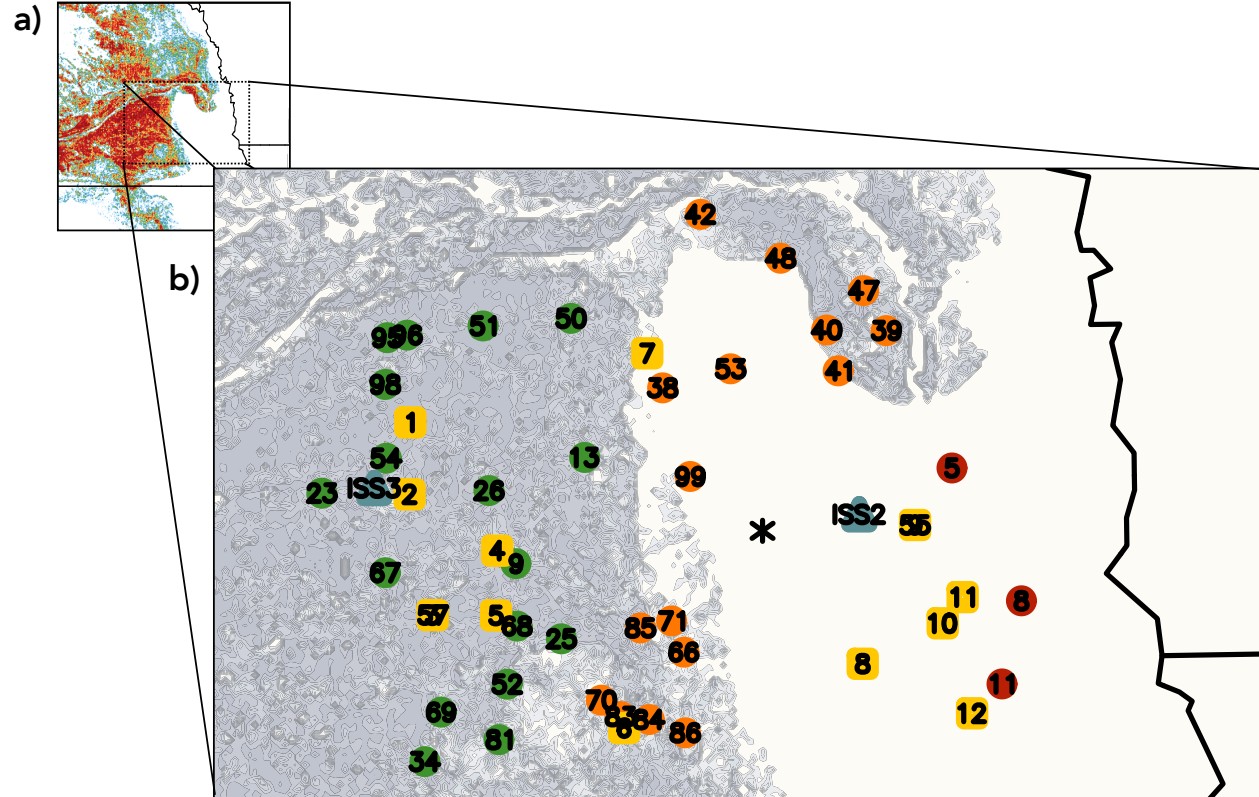

**Figure 2. a) AIM-HPA 1km irrigation dataset applied to the modeling domain with a box noting the zoomed in area in the subfigure. b) Locations of GRAINEX instruments overlaid on AIM-HPA 1km irrigation fraction dataset. Circles indicate stations with meteorological observations (i.e., temperature and humidity), classified as irrigated (green), transition (orang), and rainfed (red). Yellow squares show flux towers, blue triangles are the locations of radiosonde launches (i.e., ISS sites) at the irrigated (ISS3 - York) and non-irrigated sites (ISS2 – Roger's Farm).**






**Figure 3. Difference from control in daytime average, a-d) top layer soil moisture, e-h) evaporative fraction, i-l) temperature at 20m, m-p) humidity (mixing ratio) at 20 m, q-t) PBL height, for the AIM-HPA 1km, AIM-HPA 12km, GRIPC 1 km, and MOD 12km runs.**







Figure 4. Evaporative Fraction vs. PBL Height plots for the Irrigated subregion for the a) NO IRR, b) AIM-HPA 1km, c) AIM-HPA 12km, d) GRIPC 1km, e) MOD 12 km runs. There is one gray marker for each 1km gridcell within the 100 x 100km irrigated subregion. The colored markers represent the subregional average in each run and therefore are the same for each subplot.





**Figure 5. As in Figure 4, but for the Transition region.**






**Figure 6. Average mixing diagrams for the a) North Irrigated, b) Middle Irrigated, c) South Irrigated, d) Northeast Transition, e) Northwest Transition, f) Southwest Transition, and d) Rainfed regions.**



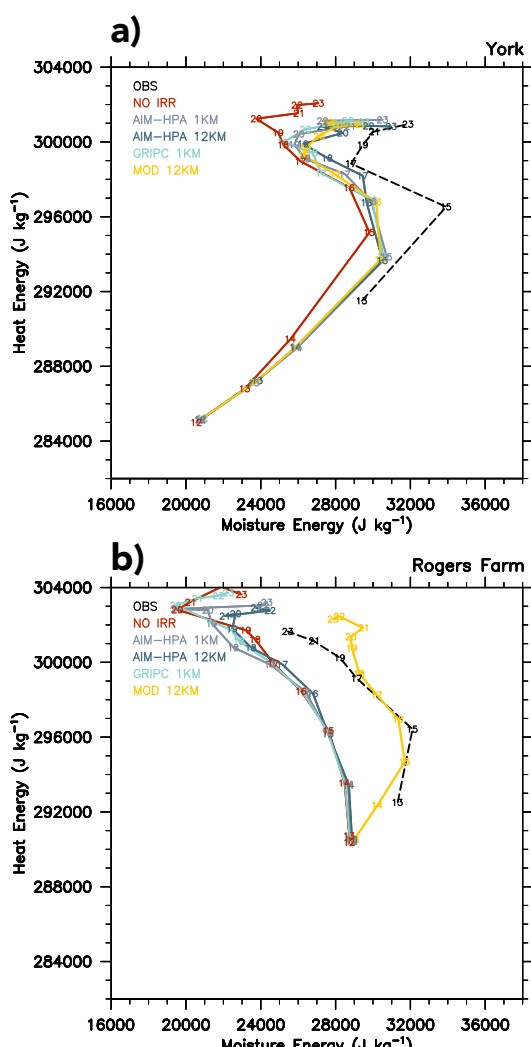

Figure 7. Mixing diagrams for the a) ISS3 York (irrigated) and b) ISS2 Roger's Farm (rainfed) sites.



**Figure 8.** Potential temperature profiles for each model run and observations at the York integrated sounding site (ISS3 - irrigated) every two hours from 15 UTC July 24 to 01 UTC July 25.




**Figure 9.** As in Figure 8 but for the Roger's Farm integrated sounding site (ISS2 - rainfed).







**Figure 10. Water vapor mixing ratio profiles for each model run and observations at the York integrated sounding site (ISS3 - irrigated) every two hours from 15 UTC July 24 to 01 UTC July 25.**



**Figure 11.** As in Figure 10 but for the Rogers Farm (Rainfed) site.
