# Peer review of "Investigating the Response of Land-Atmosphere Interactions and Feedbacks to Spatial Representation of Irrigation in a Coupled Modeling Framework"

_EGUsphere, 2023_

## Referee Comment (RC1)

This study uses a high-resolution, regional coupled modeling system to investigate the impacts of irrigation dataset selection on land-atmosphere (L-A) coupling. Results show that L-A coupling is sensitive to the choice of irrigation dataset and resolution and that the irrigation impact on surface fluxes and near surface meteorology can be dominant, conditioned on the details of the irrigation map (i.e., boundaries, heterogeneity, etc), or minimal. I'm really interested in this study. There are a few comments below.

Firstly, in my view, estimation of irrigation amount and choice of irrigation water source and irrigation method are also key aspects in parameterizing irrigation water use and modeling its impacts. So, I'm going to ask a few questions around these.

**Estimation of irrigation amount.**
1. This study chose 50% of field capacity as the irrigation trigger threshold and 80% of field capacity as the target, I am interested in why this study chose this parameter (default parameter or based on observations of irrigation amount?).
2. The immediate effect of different irrigation maps is irrigation amount of the region, but this study also lacked the validation of irrigation amount. Therefore, the simulation of irrigation amount lacks credibility. Is it feasible to make research based on this?

**Choice of irrigation water source.**
    In your study, water is withdrawn from different sources or only simple source for irrigation?

**irrigation method**
    "Water was applied as precipitation (mimicking a 'sprinkler' application)". Is this set according to the actual local conditions?

Secondly, graphical abstract: It contains too many elements, and it is difficult to understand the meaning of each sub-picture without detailed captions.

Minor comments:
Line 27, "PBL" -> "planetary boundary layer (PBL)"
Line 91, "planetary boundary layer (PBL) " -> "PBL"
Line 427, "Chen F. and Avissar R.,", Whether there is a disunity in this piece?
Line 464, "——, ——, T. E. Franz…"

---

## Author Comment (AC1)

We thank the reviewer for their careful attention to our manuscript. The paper has improved substantially as a result of their thoughtful comments. Please see our responses to the comments below in blue.

Reviewer #1:

This study uses a high-resolution, regional coupled modeling system to investigate the impacts of irrigation dataset selection on land-atmosphere (L-A) coupling. Results show that L-A coupling is sensitive to the choice of irrigation dataset and resolution and that the irrigation impact on surface fluxes and near surface meteorology can be dominant, conditioned on the details of the irrigation map (i.e., boundaries, heterogeneity, etc), or minimal. I'm really interested in this study. There are a few comments below.

Firstly, in my view, estimation of irrigation amount and choice of irrigation water source and irrigation method are also key aspects in parameterizing irrigation water use and modeling its impacts. So, I'm going to ask a few questions around these.

We fully agree that irrigation amount, source, and method can be critical factors in the simulation of irrigation impacts on the water and energy cycles. In fact, our motivation for this line of research is to improve the 'where, when, and how much' questions of irrigation in our modeling systems. There are quite a few studies where we and others address these (how much and 'how') factors directly (Lawston et al. 2015, 2017; Nie 2018,2019), but to our knowledge no study has directly addressed the impacts of the often overlooked irrigation fraction maps (which impacts where, and when). In order to complete a direct assessment of the impacts of the irrigation map independently, it was necessary to control for the other irrigation scheme factors, as varying more than one factor at once would confound the results. Each of the irrigation scheme options was carefully selected based on previous research, and represent what would be our current best approach to amount, source, and method (while recognizing these could all use work in the future as well). We expand on our reasoning for the options selected in response to the questions below and have clarified in the text where appropriate.

**Estimation of irrigation amount.**

1. This study chose 50% of field capacity as the irrigation trigger threshold and 80% of field capacity as the target, I am interested in why this study chose this parameter (default parameter or based on observations of irrigation amount?).

The irrigation scheme is designed so that the irrigation water will turn on if the root zone moisture availability in the morning (6am) is drier than a user-defined percentage of field capacity (i.e., the 'threshold'). The 80% field capacity is a default value that acts to 'turn off' the irrigation when the soil moisture reaches this value. We chose 50% for

the 'on' threshold based on previous research of Ozdogan et al (2010) who used the same irrigation scheme and modeling system, where this 50% was "based on discussions with local experts in Nebraska and California, followed by trial and error." The sprinkler irrigation scheme and thresholds were also evaluated extensively in Lawston et al. 2015 for a subset of the current study area using ground-based soil moisture and local irrigation data. The method of application (sprinkler, drip, or flood) for the irrigation schemes within this modeling system was also assessed in Lawston et al. 2015 for a subset of the study area.

The text has been updated to clarify this point and include more specific justification for using this threshold (Lines 180-184 – italicized text added or changed):

"4) The root zone must be dry enough to require irrigation, as determined by root zone moisture availability that falls below a user-defined threshold of field capacity. Ozdogan et al. (2010) determined 50% of field capacity to be sufficient based on correspondence with local experts in Nebraska and California and trial and error. Due to this previous work, as well as previous assessments of the irrigation scheme and modeling system (Lawston et al. 2017), this study also uses a threshold of 50% of field capacity."

2.  The immediate effect of different irrigation maps is irrigation amount of the region, but this study also lacked the validation of irrigation amount. Therefore, the simulation of irrigation amount lacks credibility. Is it feasible to make research based on this?

Although there was a comprehensive suite of meteorological observations taken during GRAINEX, the campaign was not able to collect information about irrigation amounts due to privacy concerns (Rappin et al. 2021). Thus, detailed information about irrigation amount during this time is not available, nor was it the goal of this study. We recognize that validating 'amount' is another aspect of irrigation modeling that is a challenge and needs further study before assessing the value of a particular irrigation scheme and application. This is unfortunately a common issue for high-resolution studies that involve human practices (McDermid et al. in press).

That being said, the work presented here is a sensitivity study where the goals are to specifically assess the impact of different irrigation maps on land-atmosphere coupling in the model and how that corresponds to the coupling we are able to observe through the comprehensive land-atmosphere GRAINEX datasets, that are quite rare for irrigated regions. We attempt to make clear that the purpose is not to discern the most accurate irrigation map (Line 162) nor to make broad conclusions about real-world effects of irrigation from the modeling results alone, as those would require a more

rigorous validation of the irrigation amounts simulated. Rather, we focus on the range of possible outcomes (in terms of land-atmosphere coupling) that one could produce from simply changing one input to the model (i.e., the irrigation fraction map), with the understanding that no single run is necessarily the 'correct' simulation. This study and its results serve as a cautionary tale to future irrigation modeling efforts that may overlook the importance of the irrigation map in the interpretation of the results.

Lines 162-165 have been revised as follows to make clear that this is a model sensitivity study:

"It should be noted that the purpose of this study is not to discern the most accurate irrigation map for the study area. Rather, this work is a model sensitivity study that seeks to understand if and to what extent irrigation heterogeneity (via irrigation map selection and resolution) can impact simulation and prediction of land-atmosphere coupling and ambient weather, and discusses the implications of such impacts in the context of sub-grid scale process representation in coarser scale models."

**Choice of irrigation water source.**

In your study, water is withdrawn from different sources or only simple source for irrigation?

In this version of the model and irrigation scheme, the irrigation water is not drawn from any particular source (e.g., groundwater, surface water, etc). We acknowledge that constraining and tracking irrigation water through the earth system is important at longer time scales and for water resources applications. There is ongoing work that connects these schemes to groundwater (Nie et al. 2019) and future work by our modeling groups will connect irrigation schemes to surface water and other management types (such as dams, etc). Our research questions, however, specifically target land-atmosphere interactions at the diurnal and local scale for which the importance of irrigation source attribution is negligible.

**irrigation method**

"Water was applied as precipitation (mimicking a 'sprinkler' application)". Is this set according to the actual local conditions?

Yes – according to the USDA, sprinkler system (e.g., center pivot systems) are the most common type of irrigation method in Nebraska, irrigating about 80% of farm acres on 68% of farms (NASS, 2009; Lawston et al. 2015).

The follow text has been added (Lines 186-187):

"Center pivot sprinklers are the most common method of irrigation in Nebraska (NASS 2009)."

Secondly, graphical abstract: It contains too many elements, and it is difficult to understand the meaning of each sub-picture without detailed captions.

We can't seem to find the graphical abstract in the manuscript documents. Perhaps Figure 1 was used for a graphical abstract by default? We would be happy to remove or revise this if directed (logistically) to where it is and how to change this.

Minor comments:
Line 27, "PBL" -> "planetary boundary layer (PBL)"

Corrected

Line 91, "planetary boundary layer (PBL) " -> "PBL"

Corrected

Line 427, "Chen F. and Avissar R.,", Whether there is a disunity in this piece?  Line 464, "——, ——, T. E. Franz..."

The entire references section has been revised to comply with the journal format.

References:
Lawston, P. M., Santanello , J. A., Jr., Zaitchik, B. F., & Rodell, M., 2015: Impact of Irrigation Methods on Land Surface Model Spinup and Initialization of WRF Forecasts, Journal of Hydrometeorology, 16(3), 1135-1154.
https://journals.ametsoc.org/view/journals/hydr/16/3/jhm-d-14-0203_1.xml

Lawston, P.M., Santanello, J. A.,, T. E. Franz, and M. Rodell, 2017: Assessment of irrigation physics in a land surface modeling framework using non-traditional and human-practice datasets. Hydrol. Earth Syst. Sci., 21, 2953–2966, https://doi.org/10.5194/hess-21-2953-2017.

Nie, W., Zaitchik, B. F., Rodell, M., Kumar, S. V., Anderson, M. C., & Hain, C. (2018). Groundwater withdrawals under drought: Reconciling GRACE and Land Surface Models in the United States High Plains Aquifer. Water Resources Research, 54, 5282–5299. https://doi.org/10.1029/2US govt work017WR022178

Nie, W., Zaitchik, B. F., Rodell, M.,Kumar, S. V., Arsenault, K. R., Li, B., &Getirana, A. (2019). AssimilatingGRACE into a land surface model inthe presence of an irrigation-induced groundwater trend. Water Resources Research, 55, https://doi.org/10.1029/2019WR025363

Sonali McDermid, S., Nocco, M., Lawston-Parker, P., Keune, J., Pokhrel, Y., Jain, M., Jägermeyr, J., Brocca, L., Massari, C., Jones, A., Vahmani, P., Thiery, W., Yao, Y., Bell, A., Chen, L., Dorigo,W., Hanasaki, N., Jasechko, S., Lo, M.-H., Mahmood, R., Mishra, V., Mueller, N. D., Dev Niyogi, D., Rabin, S., Sloat, L., Wada, Y., Zappa, L., Fei Chen, F., Cook, B. I., Kim, H., Lombardozzi, D., Polcher, J., Ryu, D., Santanello, J., Satoh, Y., Seneviratne, S., Singh, D., and Yokohata, T.: The Impacts of Irrigation in the Earth System, Nature Reviews Earth and Environment. In press.

---

## Author Comment (AC2)

We thank the reviewer for their careful attention to our manuscript. The paper has improved substantially as a result of their thoughtful comments. Please see our responses to the comments below in blue.

**Reviewer #2:**

This paper comprehensively investigates the impact of irrigation on land-atmosphere coupling using the Noah LSM, which has great scientific significance. It focuses specifically on the impact of various satellite-based irrigation products on LA coupling, and the influence of different spatial resolutions on these effects. This paper also explored the "scale effect" across multiple spatial resolutions through ground-based observations and model simulations. However, in the current version, major revisions still need to be made to the structure of the paper, and the quality of the figures also needs improvement. Here are my suggestions:

1. "Irrigation threshold" is a critical parameter that affects the irrigation timing and amount in the model simulations. However, it seems that there is no detailed discussion of this threshold in the paper. I suggest that the author provide a detailed explanation of this threshold and supplement some sensitivity analyses of the threshold selection on the results.

The reviewer is correct about the importance of the irrigation threshold on the overall behavior of the irrigation scheme (along with irrigation type, amount) and echoes the comments of Reviewer #1. As a result of this feedback, the text has been revised to include more explanation and justification of the irrigation triggering and thresholds. Please see the response to Reviewer #1, comment #1 for specific changes in this regard.

In our view, additional sensitivity analysis is not necessary, as it has already been extensively evaluated in two previous studies using this irrigation scheme and modeling system. In particular, Lawston et al. 2015 conducted a sensitivity analysis using different irrigation methods (flood, drip, and sprinkler) and different thresholds for an area of Nebraska that overlaps the current study area. In addition, Lawston et al. 2017 evaluated the impact of landcover and greenness vegetation fraction datasets on the irrigation amount and timing for a subset of the domain in this work.

The goal of this paper is to focus on the impacts to L-A coupling driven specifically by the irrigation map, as advances in computational tools and efficiencies have only very recently resulted in several choices for irrigation maps in some regions. In order to isolate the impacts of the irrigation map on the results, all other aspects of the

irrigation scheme must be held consistent. Each of these options was thoughtfully selected based on extensive previous research, detailed above.

In addition to the changes detailed in the response to Reviewer #1 (comment #1), we have also revised Lines 187-190 to clarify where additional information and sensitivity analysis can be found:

"More details about the irrigation schemes as well as an evaluation and sensitivity analysis of the irrigation scheme and thresholds can be found in Ozdogan et al. (2010) and Lawston et al. (2015, 2017). The irrigation scheme, thresholds, and all datasets except irrigation fraction (i.e., land cover, GVF, soil texture, crop type, meteorological forcing) are kept consistent between runs in order to isolate the spatial irrigation representation impact."

2. The second section (Background) and Introduction contain duplicated information. I suggest merging them to streamline this section. The current version makes it difficult for readers to capture the key messages. In addition, the description of the observation data in the paper is too brief. Due to the ground-based observations are crucial for the validation of the simulations, the authors should strengthen this part.

More information was added to the Methods section describing the observations. Lines 235-247 now read:

"Observations from the GRAINEX field campaign are used to assess the model simulations. Figure 2 shows the locations of the comprehensive land and PBL profiling instruments used in this study, overlaid on irrigation fraction given by the AIM-HPA dataset. The green, orange, and red circles in Fig. 2 note the locations of 38 Environmental Monitoring Economical Sensor Hub (EMESH) meteorological stations. EMESH weather stations were developed at the University of Alabama Huntsville and were field tested for accuracy and reliability. Each EMESH station recorded standard meteorological data such as air temperature, barometric pressure, relative humidity, wind speed and direction, and rainfall, as well as soil moisture and temperature. The blue triangles in Fig. 2 indicate the locations of two Integrated Sounding System (ISS) sites. The western ISS site (i.e., York) is surrounded by irrigated agriculture and the eastern site (i.e., Rogers Farm) is representative of the nonirrigated region. Instrumentation at each ISS site included a ceilometer, radar wind profiler, weather station, and 2 hourly radiosonde launches from sunrise (~11 UTC) to sunset (~1 UTC). In this study, we use weather data (e.g., temperature, humidity, pressure) from the EMESH stations and radiosonde observations from the Rogers Farm and York Integrated Sounding System (ISS) sites. More information about the EMESH stations

and the ISS sites, as well as a full description of all instruments deployed during the campaign can be found in Rappin et al. 2021."

In addition, information about the field campaign from which the observations were derived has been removed from the Background and placed in the Methods section (lines 223-232) to both improve the description of the observations and to streamline the Background, as the reviewer suggested.

The complex nature of this work and the fact that it spans several different communities (i.e., land heterogeneity, local land-atmosphere interactions, and irrigation) requires sufficient explanatory background and motivation to give appropriate context for the work. As a result, we have attempted to use the Introduction to set up the motivation and then expand on such points in the Background. We are open to considering additional revision of the Introduction and Background if the reviewer has specific suggestions for material that is unnecessary or duplicative.

3. The definition of "transition region" needs to be highlighted in the Method section.

The Methods section has been updated to provide more information about the definition of regions and the classification of sites. Lines 249-271 have been added:

"EMESH stations with longitude less than (i.e., west of ) 97.084°W are well-within the irrigated area and are classified as 'irrigated' stations, while those with longitude greater then (i.e., east of) 96.335°W longitude are classified as rainfed stations. The stations located between 97.084°W and 96.335°W are classified as transition stations, as they are likely subject to both irrigated and non-irrigated effects under typical synoptic conditions. These longitude cut-offs were chosen to encompass both the boundary of irrigation as given by the AIM-HPA 1km map, as well as the Big Blue River, locally understood to be the unofficial 'dividing line' between predominately irrigated and rainfed agriculture (Rappin et al. 2021). In addition, a coarser-scale subregional analysis is completed that imposes three 100 x 100 km boxes on the study region (shown in Fig.1), as proxies for three ESM gridcells that are mostly irrigated, partially irrigated (i.e., transition), and mostly rainfed, discussed in Section 4."

**Minor comments:**

**Line 24-26:** Please provide specific quantitative information instead of qualitative expressions.

This sentence was revised to include more specific information:

"A consistent finding across several analyses was that even a low percentage of irrigation fraction (i.e., <4-16%) can have significant local and downstream atmospheric impacts (e.g., lower PBL height), suggesting that representation of boundaries and heterogeneous areas within irrigated regions is particularly important for the modeling of irrigation impacts on the atmosphere in this model."

**Line 28:** Please provide the full name when PBL first appears.

Done.

**Line 33-34:** This sentence is exactly the same as the abstract, please rephrase.

The first line of the abstract has been revised so it is no longer the same as this sentence.

**Line 60:** More references.

More references have been added to this sentence.

**Line 68:** "L-A" has already been abbreviated before, so it is not necessary to provide the full name here.

The full name has been removed.

**Line 72:** What does "Irrigation Dataset" specifically refer to? Is it the irrigation fraction map or the irrigation water use map?

This sentence has been revised to: "The main questions this work seeks to answer are 1) What is the impact of irrigation dataset (i.e., irrigation fraction map) selection on land surface heterogeneity in soil moisture and surface fluxes?"

**Line 159:** Why not use satellite-based LAI?

The NCEP GVF and LAI datasets are climatological averages based on satellite data from the AVHRR satellite. In some instances, time-varying datasets (rather than climatological) can be beneficial – for example a case study featuring a drought or a case study in the Spring in a year with early greening. However, using these datasets is more computationally expensive and would likely provide minimal benefit in this case study as it is in mid-summer in a year with fairly normal temperature and precipitation. This means the climatological dataset is expected to be representative of the vegetation conditions. Regardless, had we used a time-varying LAI product in this study, it may have changed the large-scale heterogeneity but would do so in a uniform way

across all runs and therefore should not impact the conclusions presented here that are focused on differences between runs due to the irrigation map.

There is no direct dependence of the irrigation scheme on the LAI dataset. However, the irrigation scheme does use GVF to scale the root depth and irrigation amount. In a previous study, Lawston et al. 2017, the authors assessed time-varying GVF from the VIIRS satellite as compared to the climatological GVF and subsequent impacts on irrigation triggering and amounts. As mentioned in comment #1, each of the options that were kept consistent across runs (i.e., thresholds, datasets) were thoughtfully selected based on previous research, and in order to isolate the irrigation map impact.

**Line 165:** When considering irrigation as a special form of precipitation in the model, have the effects of canopy interception been taken into account? Large and lush leaves often result in more actual water consumption for irrigation, and has the cooling effect caused by the interception evaporation been calculated in the model?

Yes, the model adds the water as precipitation, which means that the irrigation water is subject to all of the same processes in the land model that a typical raindrop would undergo, including canopy interception, canopy water evaporation, and direct soil evaporation.

**Line 258-260:** Can the changes in EF be quantified specifically? This can help compare the changes in EF caused by different irrigation fraction products and the results caused by different resolutions.

This sentence has been revised as follows:

"For example, the AIM-HPA 1km dataset produces the most spatial variability in EF (i.e., numerous points spread between 0.3 and 0.7 EF across the 100km box), due to the local heterogeneity that extends to lower and, importantly, zero irrigation fraction values. This variability in EF is lost when upscaling to 12km (Fig. c)."

We focus on the magnitude of EF and PBLH in Figures 4 and 5 (rather than differences from NO-IRR). We completed an additional analysis, based on the reviewer's comment, that shows the same type of figure but differences (from NO IRR) for each run and each region. These three figures were added to the Supplement (Supplement Figures 3-5). This analysis is consistent with the previous findings and does not change the conclusions presented in the paper. For a spatial map of the change in daytime average evaporative fraction between each run and control, please see Figure 3 e-h.

**Line 308:** It is necessary to emphasize why similar data analysis needs to be conducted on individual stations.

The spatially distributed meteorological sites (i.e., near-surface temperature and humidity observations) used in the previous analyses make it possible to understand the bulk response of the atmosphere to irrigation in the irrigated, transition, and rainfed regions. However, observations of the PBL height and composition, given by the atmospheric profiles in (Figs 8-12) can illuminate features of L-A coupling and feedbacks that are otherwise unable to be gleaned from near-surface observations alone. It is difficult (i.e., time-intensive and expensive) to collect these types of PBL observations, so it is a rare to have two PBL sounding sites in a small study domain that we can take advantage of for a L-A coupling (LoCo process chain) analysis. Analyzing these two stations, (one on the irrigated side and the other on the rainfed side) first at the near-surface through mixing diagrams (Fig. 7), and then together with the potential temperature and moisture profiles (Figs 8-12), allows for a holistic understanding connecting surface and near-surface properties to those in the PBL and the lower troposphere.

**Figure 2:** A legend should be provided to explain the meaning of different colors and numbers, rather than just describe them in the figure caption.

This figure has been revised. A legend has been added, as recommended by the reviewer. Please note the yellow boxes on the previous version of this figure have been removed, as they indicated the location of flux towers that were not used in this study.

**Figure 6:** The proportion of text and numbers in Figure 6 and subsequent figures doesn't look very good, and the font size is too small to read clearly. Also, the unit of the Y-axis can be converted to MJ or displayed using scientific notation.

Many changes have been made to Figure 6 to increase readability and improve the aesthetics of the figure, including but not limited to, increasing the font size, converting the Y-axis to kJ, creating a common legend, and decreasing the x and y axes ranges. In addition, the line colors were changed to create consistency across Figures 6-11 (i.e., each run is the same color in each figure) and the line style was changed so that each run can be identified solely by the line dash pattern (rather than relying on color) to increase accessibility for colorblind readers. Many of these changes have been applied to Figures 7-11 as well.

Small changes have also been made to Figures 4 and 5. The label font size was increased to improve readability and the colored markers (previously all circles) have been changed to different markers for each run to increase accessibility for colorblind readers.

**Figure 7:** The two sub-figures in Figure 7 need to be rearranged horizontally instead of vertically.

The subfigures have been rearranged horizontally. In addition, similar revisions to Figure 6 have been made to Figure 7 (and subsequent figures) in terms of font size and aesthetics.